**Organosulfates in atmospheric aerosol: synthesis and quantitative analysis of**
**PM$_{2.5}$ from Xi'an, Northwest China**
Ru-Jin Huang[1,2,*], Junji Cao[1], Yang Chen[3], Lu Yang[1], Jincan Shen[4], Qihua You[2], Kai Wang[1,5],
Chunshui Lin[1,6], Wei Xu[1,6], Bo Gao[1], Yongjie Li[7], Qi Chen[8], Thorsten Hoffmann[5], Colin D.
O'Dowd[6], Merete Bilde[9], Marianne Glasius[9]
[1]Key Laboratory of Aerosol Chemistry and Physics, State Key Laboratory of Loess and
Quaternary Geology, Institute of Earth and Environment, Chinese Academy of Sciences, Xi'an,
710061, China
[2]Centre for Atmospheric and Marine Sciences, Xiamen Huaxia University, Xiamen 361024,
China
[3]Key Laboratory of Reservoir Aquatic Environment of CAS, Chongqing Institute of Green
and Intelligent Technology, Chinese Academy of Sciences, Chongqing 400714, China
[4]Food Inspection & Quarantine Center of Shenzhen Entry-Exit Inspection and Quarantine
Bureau, Shenzhen Key Laboratory of Detection Technology R&D on Food Safety, Shenzhen
518045, China
[5]Institute of Inorganic and Analytical Chemistry, Johannes Gutenberg University of Mainz,
Duesbergweg 10–14, 55128 Mainz, Germany
[6]School of Physics and Centre for Climate and Air Pollution Studies, Ryan Institute, National
University of Ireland Galway, University Road, Galway, Ireland
[7]Department of Civil and Environmental Engineering, Faculty of Science and Technology,
University of Macau, Taipa, Macau, China
[8]State Key Joint Laboratory of Environmental Simulation and Pollution Control, College of
Environmental Sciences and Engineering, Peking University, Beijing 100871, China
[9]Department of Chemistry, Aarhus University, Langelandsgade 140, DK-8000 Aarhus C,
Denmark
*Corresponding author. E-mail address: rujin.huang@ieecas.cn; Tel: +86 (0)29 6233 6275
**Abstract**
The sources, formation mechanism and amount of organosulfates (OS) in atmospheric aerosol
are not yet well understood, partly due to the lack of authentic standards for quantification. In
this study, we report an improved robust procedure for the synthesis of organosulfates with
different functional groups. Nine authentic organosulfate standards were synthesized and four
standards (benzyl sulfate, phenyl sulfate, glycolic acid sulfate, and hydroxyacetone sulfate)
were used to quantify their ambient concentrations. The authentic standards and ambient
aerosol samples were analyzed using an optimized ultra performance liquid chromatography-
electrospray ionization-tandem mass spectrometric method (UPLC−ESI−MS/MS). The
recovery ranged from 80.4% to 93.2%, the limits of detection and limits of quantification
obtained were 1.1-16.7 pg m$^{-3}$ and 3.4-55.6 pg m$^{-3}$, respectively. Measurements of ambient
aerosol particle samples collected in winter 2013-2014 in urban Xi'an, northwest China, show
that glycolic acid sulfate (77.3 ± 49.2 ng m$^{-3}$) is the most abundant species of the identified
organosulfates followed by hydroxyacetone sulfate (1.3 ± 0.5 ng m$^{-3}$), phenyl sulfate (0.14 ±
0.09 ng m$^{-3}$), and benzyl sulfate (0.04 ± 0.01 ng m$^{-3}$). Except for hydroxyacetone sulfate, which
seems to form mainly from biogenic emissions in this region, the organosulfates quantified
during winter in Xi'an show an increasing trend with an increase in the mass concentrations of
organic carbon indicating their anthropogenic origin.

**1 Introduction**
Atmospheric aerosol particles represent a highly complex blend of inorganic and organic matter
originating from a wide variety of both natural and anthropogenic sources. The organic fraction
typically constitutes 20-90% of the total submicron aerosol mass and is much less constrained
in terms of chemical composition than the inorganic fraction (Jimenez et al., 2009; Hallquist et
al., 2009). Only ~10-30% of the particulate organic matter has been identified as specific
compounds despite years of effort and the use of the most sophisticated techniques available
(Hoffmann et al., 2011). The insufficient knowledge of the composition of organic aerosol
particles at the molecular level hinders a better understanding of the sources, formation and
atmospheric processes of organic aerosol as well as their physicochemical properties and effects
on climate and human health (Noziere et al., 2015).

Organosulfates are ubiquitous in atmospheric aerosol and have been detected in ambient aerosol
particles from America, Europe, Asia and the Arctic during the last decade (e.g. Surratt et al.,
2008; Iinuma et al., 2007; Stone et al., 2012; Hansen et al., 2014; Kourtchev et al., 2016; Surratt
et al., 2007). Due to the presence of the deprotonated functional group $R-O-SO_3^-$,
organosulfates are acidic and highly water soluble and therefore can enhance the aerosol
hygroscopicity. These characteristics, together with the light-absorbing property of
organosulfates, lead to potential impacts on climate (Lin et al., 2014).

Organosulfates are tracers of secondary organic aerosol (SOA) formation and have been
demonstrated to be produced from heterogeneous and multiphase reactions (e.g. Surratt et al.,
2008; Iinuma et al., 2007; Chan et al., 2011; Zhang et al., 2012). Chamber studies have found
that the oxidation of biogenic volatile organic compounds (BVOCs) including isoprene,
monoterpenes, and sesquiterpenes can form organosulfates on acidified sulfate particles (e.g.
Surratt et al., 2008; Iinuma et al., 2007; Chan et al., 2011; Zhang et al., 2012). A very recent
study revealed a previously unrecognized pathway for organosulfate formation through the
heterogeneous reaction of $SO_2$ with the unsaturated bond in oleic acid (Shang et al., 2016). A
number of biogenic organosulfates have been observed in ambient aerosol, in particular,
isoprene-derived organosulfates (e.g. Kristensen et al., 2011; He et al., 2014; Liao et al., 2015;
Budisulistiorini et al., 2015). A recent study reported the formation of aromatic organosulfates
by photochemical oxidation of polycyclic aromatic hydrocarbons (PAHs) in the presence of
sulfate seed particles (Riva et al., 2016). Aromatic organosulfates have also recently been
observed in urban aerosol from different locations in Asia. The presence of aromatic
organosulfates was first suggested by Stone et al. (Stone et al., 2012) based on analysis of
aerosol samples collected at four sites in Asia. Kundu et al. (Kundu et al., 2013) quantified
benzyl sulfate ($C_7H_7SO_4^-$) and identified its homologous series with increasing number of
methylene groups ($C_8H_9SO_4^-$ and $C_9H_{11}SO_4^-$) in Lahore, Pakistan. Furthermore, Staudt et al.
(Staudt et al., 2014) synthesized phenyl sulfate, benzyl sulfate, 3- and 4-methylphenyl sulfate

and 2-, 3-, and 4-methylbenzyl sulfate and quantified them in aerosols collected in urban samples from Lahore and Pasadena, USA as well as Nepal. Ma et al. (Ma et al., 2014) reported the contribution up to 64% from aromatic organosulfates to the sum of identified organosulfates in winter Shanghai, while Wang et al. (Wang et al., 2016) found aromatic organosulfates to constitute less than 22% of the detected number of organosulfates in Shanghai, Nanjing, and Wuhan.

Organosulfates have been estimated to contribute 5-10% of the organic mass in fine particles in the USA (Tolocka and Turpin, 2012). However, quantification of organosulfates is a challenging task due to the lack of authentic standards and incomplete understanding of the sources, precursors and formation processes of organosulfates. To date, many studies of organosulfates have remained at the qualitative level, although a limited number of studies have provided quantitative or semi-quantitative analysis of certain organosulfates (e.g. Kundu et al., 2013; Staudt et al., 2014; Ma et al., 2014; Olson et al., 2011; Hettiyadura et al., 2017). Moreover, several studies show that organosulfates are present as a wide range of species with individual species such as the organosulfate derived from isoprene epoxydiols (IEPOX) contributing 0.2-1.4% of the total organic aerosol mass (Liao et al., 2015). This further complicates the quantification of organosulfates. A few organosulfate standards have been synthesized for quantification purposes. For example, Olsen et al. (Olson et al., 2011) measured 0.4-3.8 ng m$^{-3}$ lactic acid sulfate and 1.9-11.3 ng m$^{-3}$ glycolic acid sulfate in samples of PM$_{2.5}$ (particulate matter with an aerodynamic diameter <2.5 μm) from the US, Mexico City, and Pakistan. Kundu et al. (Kundu et al., 2013) measured monthly-average concentrations of benzyl sulfate ranging from 0.05 to 0.5 ng m$^{-3}$ in PM$_{2.5}$ samples from Lahore, Pakistan. Staudt et al. (Staudt et al., 2014) quantified benzyl sulfate ranging from 4 to 90 pg m$^{-3}$ in PM$_{2.5}$ samples from Lahore (Pakistan), Godavari (Nepal), and Pasadena (California), while methylbenzyl sulfates, phenyl sulfate, and methylphenyl sulfates were observed intermittently in these three locations. Furthermore, Hettiyadura et al. (Hettiyadura et al., 2015) developed a hydrophilic interaction liquid chromatography method using an amide stationary phase providing excellent retention of carboxy-organosulfates and isoprene-derived organosulfates, which was validated using six model organosulfates including aliphatic and aromatic organosulfates.

Previous field studies focusing on organosulfates were conducted mainly in Europe (e.g. Iinuma et al., 2007; Kristensen et al., 2011; Gómez-González et al., 2008; Gómez-González et al., 2012; Nguyen et al., 2014; Martinsson et al., 2017) and North America (e.g. Surratt et al., 2007; Nguyen et al., 2012; Worton et al., 2011), and only a few in China (He et al., 2014; Ma et al., 2014). The particulate air pollution has been a serious environmental problem during recent winters in China, characterized by high secondary aerosol concentrations including sulfate and SOA (e.g. Huang et al., 2014; Elser et al., 2016; Wang et al., 2017). As organosulfates are tracers for SOA, more studies on organosulfates will help to better understand and constrain the SOA formation mechanisms in highly polluted regions (e.g., China) and to reconcile the underestimation of particle-phase organic carbon in atmospheric models.

In this study, nine organosulfate standards (phenyl sulfate, 3-methylphenyl sulfate, benzyl sulfate, 2-methyl benzyl sulfate, 3-methyl benzyl sulfate, 2, 4-dimethyl benzyl sulfate, 3, 5-

dimethyl benzyl sulfate, hydroxyacetone sulfate, and glycolic acid sulfate) were synthesized using an approach modified from Staudt et al. (Staudt et al., 2014) and Hettiyadura et al. (Hettiyadura et al., 2015). These authentic standards were used to optimize an ultra performance liquid chromatography electrospray ionization-tandem mass spectrometric method (UPLC–ESI–MS/MS) for the quantification of organosulfates. The presence and concentration of four of these organosulfates, namely, benzyl sulfate, phenyl sulfate, glycolic acid sulfate, and hydroxyacetone sulfate, were determined in ambient $PM_{2.5}$ collected in urban air in Xi'an, China. The rest five organosulfates were not quantified in ambient $PM_{2.5}$ because the standards were synthesized at a later stage of the study.

**2 Material and methods**

2.1 Chemicals and synthesis

The chemicals used for the synthesis of organosulfates included hydroxyacetone (99%, Sigma Aldrich), glycolic acid (99%, Sigma Aldrich), phenol (99.5%, Tic), benzyl alcohol (99.8%, Aladdin, Shanghai, China), m-cresol (99%, Sigma Aldrich), sulfur trioxide pyridine complex (98%, Sigma Aldrich), pyridine (99.9%, Sigma Aldrich), Dowex® 50WX8 (hydrogen form, 100-200 mesh, Sigma Aldrich). MilliQ water (18.2 MΩ) was used, and all other reagents were analytical grade and used without further purification.

The organosulfate standards were synthesized using a general approach modified from Staudt et al. (Staudt et al., 2014) and Hettiyadura et al. (Hettiyadura et al., 2015). Fig. 1 shows the reaction scheme. In general, alcohol (7.0 mmol) and sulfur trioxide pyridine complex (1.2 equiv.) was dissolved in dry pyridine (10 mL) in an oven-dried, three-necked flask provided with magnetic stirring under nitrogen. The reaction mixture was stirred at 30 °C for 24 h, and then the solvent was removed via distillation under vacuum at 50 °C. The residue was redissolved in distilled water (10 mL) and titrated with 0.9 M KOH until pH was above 12. Neat ethanol (40 mL, 65 °C) was added to the aqueous solution. The resulting solution was heated to reflux followed by a quick vacuum filtration to remove the stark white precipitate. The mother liquor was then placed in a freezer (-25 °C) overnight. The potassium salts of organosulfate formed in the mother liquor were collected by vacuum filtration, rinsed with cold ethanol three times and dried to obtain the target product. The synthesized organosulfate standards were stored in refrigerator (∼4 °C) and no decomposition was observed after 2 years as confirmed by nuclear magnetic resonance (NMR) analysis.

2.2 Characterization

The synthesized products were characterized with NMR and ESI–MS. [1]H NMR and [13]C NMR spectra were recorded on a Bruker Advance-III 400 MHz spectrometer at 400 and 100 MHz, respectively using trimethylsilane (TMS) as an internal standard. Chemical shifts are reported in ppm downfield from the internal reference. The NMR spectra are shown in Supplementary Information. The following abbreviations are used for the multiplicities: s = singlet, m = multiplet. The yield for phenyl sulfate was 45%, [1]H NMR (400 MHz, $D_2O$): δ/ppm 7.29-7.43 (m, 5H), [13]C NMR (100 MHz, $D_2O$): δ/ppm 121.6, 126.4, 129.8, 151.2. The yield for benzyl sulfate was 70%, [1]H NMR (400 MHz, DMSO-$d$6): δ/ppm 7.25-7.40 (m, 5 H), 4.76 (s, 2 H), [13]C NMR (100 MHz, DMSO-$d$6): δ/ppm 67.9, 127.8, 128.0, 128.6, 138.4. The yield for

hydroxyacetone sulfate was 45%, [1]H NMR (400 MHz, DMSO-*d*6): δ/ppm 4.22 (s, 2 H), 2.11
(s, 3 H), [13]C NMR (100 MHz, DMSO-*d*6): δ/ppm 26.9, 71.4, 207.0. The yield for glycolic acid
sulfate was 35%, [1]H NMR (400 MHz, DMSO-*d*6): δ/ppm 4.07 (s, 2H), [13]C NMR (100 MHz,
DMSO-*d*6): δ/ppm 65.0, 173.1. The organosulfate standards were recrystallized in ethanol
for purification and purity of these synthesized standards is >95%, confirmed by NMR
analysis. Exact mass spectra were recorded on a high-resolution mass spectrometer (HR−MS,
Q Exactive Plus, Thermo Scientific, USA) equipped with an ESI source in the negative ion
mode (ESI-). The ESI conditions were as follows: spray voltage -3.2 kV, collision energy (CE)
40 V for benzyl sulfate and 45 V for hydroxyacetone sulfate, 3-methylphenyl sulfate, glycolic
acid sulfate and phenyl sulfate, capillary temperature 350 °C, aux gas heater temperature 320 °C,
sheath gas flow rate 35, aux gas flow rate 10. The mass resolving power was 70,000. Data
acquisition was performed with *m/z* ranging from 50 to 200.

2.3 PM$_{2.5}$ samples
The 24-h integrated PM$_{2.5}$ samples were collected on pre-baked (780 °C, 3 h) quartz-fiber filters
(8×10 inch, Whatman, QM-A, USA) using a high-volume sampler (Tisch, Cleveland, OH, USA)
at a flow rate of 1.05 m$^3$ min$^{-1}$ from 18 December 2013 to 17 February 2014. After collection,
the filter samples were immediately wrapped in pre-baked aluminum foil and stored in a freezer
(below –20 °C) until analysis. The sampling site was located on the rooftop of the Institute of
Earth and Environment (~10 m above the ground), Chinese Academy of Sciences (IEECAS,
34.23°N, 108.88°E), which is surrounded by residential, commercial and trafficked areas.

2.4 Sample analysis
A portion of the filter (6 × 0.526 cm$^2$ punch) taken from each sample was sonicated for 25 min
in 9 mL of acetonitrile (ACN)/water mixture (95:5, V/V). The extracts were filtered through a
0.22 *μ*m polypropylene membrane syringe filter to remove insoluble material. The eluate was
concentrated almost to dryness with a gentle stream of purified nitrogen (99.999%) at 45 °C
using an evaporation system (TurboVap® LV, biotage), then redissolved in 500 μL of
acetonitrile/water mixture (V/V, 95:5). The prepared samples were stored at 4 °C in the
refrigerator and analyzed within 24 h. The separation and quantification were realized using a
ACQUITY UPLC system (equipped with a quaternary pump, autosampler, and thermostated
column compartment) coupled to a tandem mass spectrometer (Xevo TQ MS, Waters, USA).
The separation was carried out using a BEH amide column (2.1mm×100 mm, 1.7 μm particle
size, Waters, USA) equipped with a pre-column. The column was maintained at 35 °C and the
flow rate of mobile phase was 0.25 mL min$^{-1}$. A 5 μL injection volume was used for quantitative
analysis of samples and standards. The optimized mobile phase A (organic) consisted of
ammonium acetate buffer (5 mM, pH 8.5) in ACN and ultra-pure water (95:5, V/V) and mobile
phase B (aqueous) consisted of ammonium acetate buffer (5 mM, pH 9) in ultra-pure water. A
mobile phase gradient was used: mobile phase A was maintained at 98% for 2 min, then
decreased to 60% from 2 to 5 min and then held there for 2 min; from 7 to 12 min mobile phase
A was returned to 98%. Organosulfates were detected by a TQ MS equipped with an ESI source
in the negative ion mode. The mass spectrometer was operated in multiple reaction monitoring
(MRM) mode. Optimized MS conditions for the four organosulfates chosen for the field studies
(e.g., cone voltages and collision energies) are listed in Table 1. The capillary voltage was 2.7
kV, source temperature was 150 °C, desolvation temperature was 350 °C, desolvation gas ($N_2$)
flow rate at 800 L h$^{-1}$, cone gas ($N_2$) flow rate was 150 L h$^{-1}$, and collision gas (Ar) flow rate
was 0.16 mL min$^{-1}$. All data were acquired and processed using MassLynx software (version
4.1). All samples and standard spectra were background subtracted.
2.5 Quality control
For every 10 analyses, a procedural blank and a spiked sample−namely, real ambient samples
spiked with known amounts of a standard solution of organosulfates to be quantified were
measured to check for interference and cross-contamination. The external standard method was
used for quantitative determination of the analytes. The limits of detection are defined as the
minimum detectable peaks of individual species with a signal-to-noise (S/N) ratio of 3:1. The
recoveries were determined by the analysis of the spiked samples: we first measured a filter
punch without spike and then measured the second punch from the identical filter spiked with
known amounts of a standard solution of organosulfates. The differences between these two
measurements were divided by the amounts of organosulfates spiked to calculate the recoveries
of individual organosulfates. This recovery test also provides an indication of potential matrix
effect. The reproducibility (relative standard deviation, RSD) was determined by measuring
five identical samples that were subjected to the same pretreatment procedure. The field blank
samples were collected and analyzed, and the data reported here were corrected for the field
blanks.
**3 Results and discussion**
3.1 Mass spectral fragmentation and UPLC separation
Each synthesized organosulfate was analyzed by high resolution tandem MS (MS/MS). The
molecular ion for each organosulfate was assigned to the deprotonated molecule (R−O−SO$_3^-$).
Major sulfur-containing product ions included the sulfite ion radical (·SO$_3^-$, $m/z$ 80) that is
formed from the homolytic cleavage of the O−S bond, the sulfate ion radical (·SO$_4^-$, $m/z$ 96)
that is formed from the homolytic cleavage of the C−O bond, the bisulfite anion (HSO$_3^-$, $m/z$
81) that is formed from hydrogen abstraction followed by the heterolytic cleavage of the O−S
bond, and the bisulfate anion (HSO$_4^-$, $m/z$ 97). Phenyl sulfate, 3-methylphenyl sulfate, and
glycolic acid sulfate produce phenoxide (C$_6$H$_5$O$^-$, $m/z$ 93), 3-methylphenoxide (C$_7$H$_7$O$^-$, $m/z$
107) and glycolate (C$_2$H$_3$O$_3^-$, $m/z$ 75) anions, respectively, formed from neutral loss of SO$_3$. The
mass spectra of these compounds are shown in Fig. 2. The mass spectrum of phenyl sulfate is
similar to that reported by Staudt et al. (Staudt et al., 2014), the mass spectra of hydroxy acetone
sulfate and glycolic acid sulfate are similar to those reported by Hettiyadura et al. (Hettiyadura
et al., 2015) and the spectrum of benzyl sulfate is similar to that reported by Kundu et al. (Kundu
et al., 2013), confirming the identity of the compounds.
The ESI-MS/MS in MRM mode is applied for the quantification of individual organosulfates.
This can greatly enhance the selectivity and sensitivity by monitoring a transition pair of
precursor and product ions and thus eliminating potential interferences from the complex
aerosol matrix. Table 1 shows the optimized ESI- conditions and the transition pairs for each
organosulfate studied. The organosulfate standards were separated by UPLC using a BEH
amide column that retains extremely polar compounds through ionic, hydrogen bonding and
dipole interactions. A gradient elution procedure was applied and the aqueous portion of the
mobile phase increased from 7-43%, leading to the baseline separation of four organosulfates
within 6 min (Fig. 3a). The retention time was 0.86 min for phenyl sulfate, 0.96 min for benzyl
sulfate, 1.10 min for hydroxyacetone sulfate and 5.78 min for glycolic acid sulfate, respectively.
The mobile phase was buffered to slightly basic pH to maintain the deprotonated state of the
organosulfates, which favors the separation. The amide functionalization of the BEH stationary
phase introduces hydrogen bonding and strengthens interaction with organosulfates particularly
for those containing carboxyl and hydroxyl functional groups. It should be noted that the
chromatographic peak-broadening occurred particularly for phenyl sulfate and hydroxyacetone
sulfate when analyzing the ambient samples (Fig. 3b). This might be explained by matrix effects
due to the complex samples, which can influence the partitioning of analyte between the
stationary phase and mobile phase, particularly for those analytes with weak retention on the
column. However, the quantification of organosulfates is not affected by the peak broadening
because the transition pair of precursor and product ions used in the MRM mode of the mass
spectrometer guarantees selectivity and accuracy.
3.2 Method validation
Table 2 shows the analytical performance of the method under optimized UPLC and MS/MS
conditions. The calibration curves of each organosulfates are highly linear ($R^2 \geq 0.995$), ranging
from 0.1-40 ng mL$^{-1}$ for phenyl sulfate and benzyl sulfate, from 0.3-120 ng mL$^{-1}$ for
hydroxyacetone sulfate, and 2.0-800 ng mL$^{-1}$ for glycolic acid sulfate. The recoveries,
determined by analyzing ambient samples spiked with known amounts of organosulfate
standards, ranging from 80.4-93.2%. The good recoveries indicate high extraction efficiency,
low sample matrix effect and low error from sample pretreatment and the UPLC-MS
measurement. The limit of detection (LOD, S/N=3) and limit of quantification (LOQ, S/N=10)
ranged from 0.03 to 0.42 ng mL$^{-1}$ and 0.09 to 1.4 ng mL$^{-1}$ of the extracts, respectively. This
corresponds to LODs of 1.1 to 16.7 pg m$^{-3}$ and LOQs of 3.4 to 55.6 pg m$^{-3}$, respectively, using
the current set-up (see experimental section).
3.3 Quantification of organosulfates in ambient aerosol
Ambient PM$_{2.5}$ samples were extracted and analyzed by UPLC-MS/MS following the same
procedure as the OS standards. The four selected organosulfates were identified according to
the transition pairs of precursor and product ions of individual compounds on the MS/MS as
well as the UPLC retention time. Table 3 shows the concentrations of phenyl sulfate, benzyl
sulfate, hydroxyacetone sulfate, and glycolic acid sulfate in PM$_{2.5}$ samples collected at Xi'an
(this work), together with concentrations reported in the literature from other locations
worldwide for comparison. In our samples from Xi'an glycolic acid sulfate (average 77.3 $\pm$
49.2 ng m$^{-3}$, range 18.1-155.5 ng m$^{-3}$) was the most abundant species of the identified
organosulfate followed by hydroxyacetone sulfate (average 1.3 $\pm$ 0.5 ng m$^{-3}$, range 0.9-2.6 ng
m$^{-3}$), phenyl sulfate (average 0.14 $\pm$ 0.09 ng m$^{-3}$, range 0.04-0.31 ng m$^{-3}$) and benzyl sulfate
(average 0.04 $\pm$ 0.01 ng m$^{-3}$, range 0.03-0.06 ng m$^{-3}$).
The concentration of glycolic acid sulfate quantified in this study is about one order of
magnitude higher than those reported in the literature (see Table 3), indicating the substantial

formation of this secondary organic compound in polluted urban Xi'an. Glycolic acid sulfate can form efficiently from glycolic acid relative to glyoxal in the presence of acidic sulfate particles (Olson et al., 2011). While both organic precursors (glycolic acid and glyoxal) have biogenic and anthropogenic origins, they form mainly from the oxidation of anthropogenic emissions during winter in Xi'an. The concentrations of particle-phase glyoxal and glycolic acid measured at Xi'an during winter have been reported to be significantly higher compared to other studied regions (e.g. Kawamura and Yasui, 2005; Miyazaki et al., 2009; Cheng et al., 2013), which therefore may explain the elevated glycolic acid sulfate. The concentration of the other three organosulfates quantified in this study was much lower, but falling into the ranges measured in other regions.

It is noted that the time series of glycolic acid sulfate, phenyl sulfate, and benzyl sulfate is similar to that of organic carbon (OC) and $SO_4^{2-}$, while the concentration of hydroxyacetone sulfate did not show an increasing trend when the concentrations of OC increased (Fig. 4a). Hydroxyacetone sulfate can form from photochemical oxidation of isoprene and/or isoprene ozonolysis in the presence of acidic sulfate aerosols (Surratt et al., 2008; Riva et al., 2015), although hydroxyacetone was also suggested to originate from anthropogenic emissions (e.g., biomass burning and fossil fuel combustion) (Hansen et al., 2014). Also, the formation rate of biogenic hydroxyacetone sulfate and anthropogenic hydroxyacetone sulfate may different. This may explain the lack of correlation between hydroxyacetone sulfate and OC during winter in Xi'an. The average concentrations of glycolic acid sulfate, phenyl sulfate, and benzyl sulfate were 1.3-3.2 times higher during high pollution days ($PM_{2.5}$ range of 293.7-314.5 $\mu g\ m^{-3}$ with an average of 300.6 $\mu g\ m^{-3}$) than during low pollution days ($PM_{2.5}$ range of 94.7-121.2 $\mu g\ m^{-3}$ with an average of 106.4 $\mu g\ m^{-3}$), while the average concentrations of hydroxyacetone sulfate were rather similar between high pollution days and low pollution days (Fig. 4b). These four organosulfates together account for 0.25% of total sulfur and 0.05% of OC, respectively.

**4 Conclusions**

Nine authentic organosulfate standards, including phenyl sulfate, 3-methylphenyl sulfate, benzyl sulfate, 2-methyl benzyl sulfate, 3-methyl benzyl sulfate, 2, 4-dimethyl benzyl sulfate, 3, 5-dimethyl benzyl sulfate, hydroxyacetone sulfate, and glycolic acid sulfate, were synthesized in this study using an improved robust procedure. The synthesized compounds of benzyl sulfate, phenyl sulfate, glycolic acid sulfate, and hydroxyacetone sulfate were used as standards for quantification of these molecules in ambient $PM_{2.5}$ samples. The other five organosulfate standards were synthesized, but not used for quantification of ambient samples in this study. An improved UPLC-ESI-MS/MS method was developed and optimized for the quantification. The recovery ranges from 80.4-93.2%, and the limits of detection and limits of quantification obtained are 1.1-16.7 $pg\ m^{-3}$ and 3.4-55.6 $pg\ m^{-3}$, respectively. Measurements of $PM_{2.5}$ samples from Xi'an show that glycolic acid sulfate (77.3 ± 49.2 $ng\ m^{-3}$) is the most abundant organosulfate followed by hydroxyacetone sulfate (1.3 ± 0.5 $ng\ m^{-3}$), phenyl sulfate (0.14 ± 0.09 $ng\ m^{-3}$), and benzyl sulfate (0.04 ± 0.01 $ng\ m^{-3}$). Glycolic acid sulfate, phenyl sulfate, and benzyl sulfate show an increasing trend with the increase of OC concentrations indicating their anthropogenic origin.

*Acknowledgements.* This work was supported by the National Natural Science Foundation of China (NSFC) under Grant No. 91644219, No. 41650110488, the Minjiang Scholar Program, and the Carlsberg Foundation.

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

Table 1. The optimized ESI-MS/MS parameters and UPLC retention time of measured
organosulfates

| Organosulfate | Deprotonate molecule (*m/z*) | Product ion (*m/z*) | Cone voltage (V) | Collision energy (eV) | Retention time (min) |
|---|---|---|---|---|---|
| Phenyl sulfate | $C_6H_5SO_4^-$ (173) | $SO_3^-$ (80) $C_6H_5O^-$ (93) | 41 | 20 21 | 0.86 ± 0.02 |
| Benzyl sulfate | $C_7H_7SO_4^-$ (187) | $HSO_3^-$ (81) $SO_4^-$ (96) | 42 | 19 22 | 0.96 ± 0.02 |
| Hydroxyacetone sulfate | $C_3H_5SO_5^-$ (153) | $SO_3^-$ (80) $HSO_4^-$ (97) | 32 | 18 20 | 1.10 ± 0.02 |
| Glycolic acid sulfate | $C_2H_3SO_6^-$ (155) | $C_2H_3O_3^-$ (75) $HSO_4^-$ (97) | 26 | 18 14 | 5.78 ± 0.03 |


Table 2. Analytical performance of the UPLC-ESI-MS/MS method for organosulfate analysis

| Organosulfate | Linear range (ng mL$^{-1}$) | Linearity ($R^2$) | Recovery % | LOD (pg), injection volume (5 μL) | LOQ (pg), injection volume (5 μL) | LOD* (pg m$^{-3}$) | LOQ* (pg m$^{-3}$) |
|---|---|---|---|---|---|---|---|
| Phenyl sulfate | 0.1-40 | 0.998 | 80.4 | 0.13 | 0.43 | 1.1 | 3.5 |
| Benzyl sulfate | 0.1-40 | 0.998 | 89.6 | 0.13 | 0.43 | 1.1 | 3.4 |
| Hydroxyacetone sulfate | 0.3-120 | 0.997 | 93.2 | 2.1 | 6.9 | 16.7 | 55.6 |
| Glycolic acid sulfate | 2-800 | 0.995 | 92.0 | 0.27 | 0.88 | 2.1 | 7.1 |

*For analyzing 6×0.526 cm punches of filters collected with high-volume samplers (sampling
at 1.13 m$^3$ min$^{-1}$ for 24 h on 8"×10" filters).

Table 3. The quantification of organosulfates at Xi'an and comparison with data reported in the
literature

| Location | Date | PM$_{2.5}$ µg m$^{-3}$ | OC µg m$^{-3}$ | Organosulfate ng m$^{-3}$ | | | | Ref. |
|---|---|---|---|---|---|---|---|---|
| | | | | benzyl sulfate | phenyl sulfate | hydroxyacetone sulfate | glycolic acid sulfate | |
| Riverside, CA | 27/07/05 | 16.5 | 7.6 | - | - | - | 3.3 | Olson et al., 2011 |
| Mexico City (T0) | 26/03/06 | 40 | 8.5 | - | - | - | 4.1 | |
| Mexico City (T1) | 26/03/06 | 33 | 5.2 | - | - | - | 7.0 | |
| Cleveland, OH | 15/07/07 | 12.7 | 3.9 | - | - | - | 1.9 | |
| Bakersfield, CA | 16-18/06/10 | 11.1-12.0 | 4.0-4.8 | - | - | - | 4.5-5.4 | |
| Lahore, Pakistan | 02/11/07 | 327.5 | 174.7 | - | - | - | 11.3 | |
| Lahore, Pakistan | 12/01/2007 - 13/01/2008 | - | - | 0.05-0.50 | - | - | - | Kundu et al., 2013 |
| Lahore, Pakistan | March/07 | 177.1 | 44.6 | 0.09 | 0.004 | - | - | Staudt et al., 2014 |
| Godavari, Nepal | Feb/07 | 42.0 | 4.7 | 0.004 | ND | - | - | |
| Pasadena, CA | 5-6/06/10 | 41.8-44.1 | 7.3-7.6 | 0.006-0.007 | ND | - | - | |
| Centreville, AL | 10-11/07/13 | - | - | ND | ND | 2.7-5.8 | 9-14 | Hettiyadura et al., 2015 |
| Shanghai | 5-7/04/12 12-14/07/12 27-29/10/12 14-16/01/13 | - | - | 0.3-0.8 | - | - | - | Ma et al., 2014 |
| Xi'an (n=10) | 18/12/13-17/02/14 | 94.7-314.5 | 14.9-68.5 | 0.03-0.06 | 0.04-0.31 | 0.9-2.6 | 18.1-155.5 | This study |



Figure 1. General scheme for the synthesis of organosulfate, modified from Staudt et al.
(Staudt et al., 2014) and Hettiyadura et al. (Hettiyadura et al., 2015).

















(a)

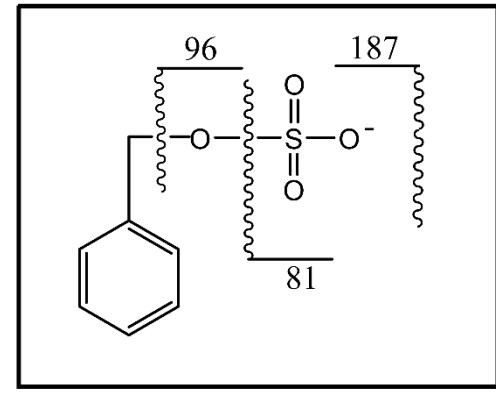 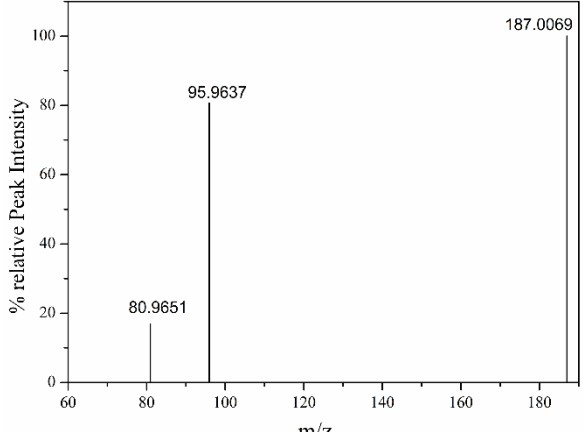

(b)

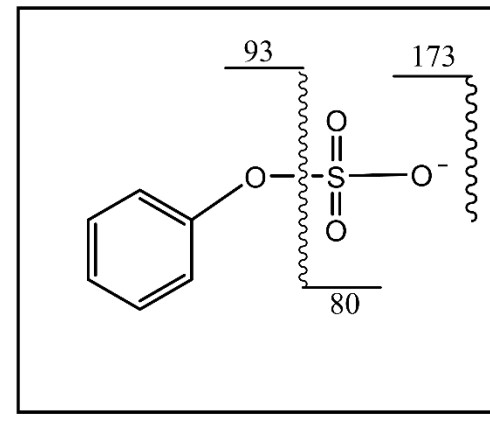 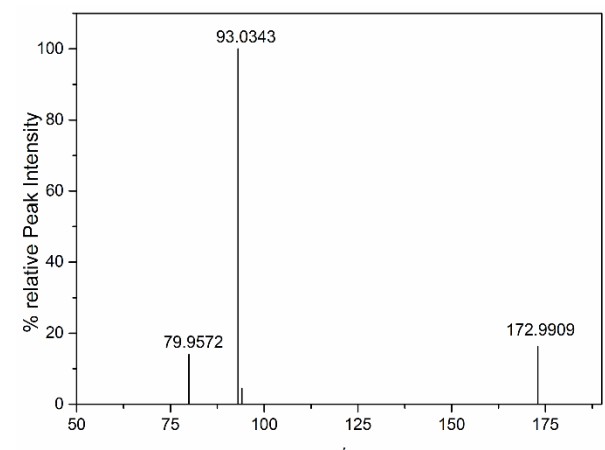

(c)

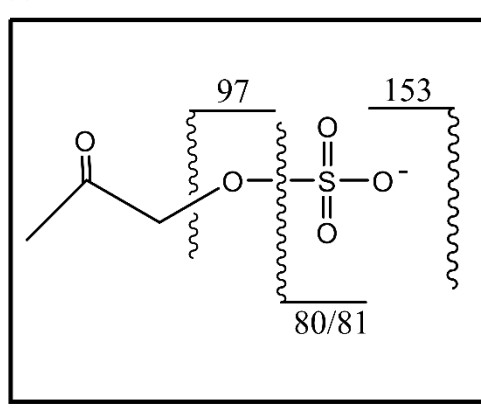 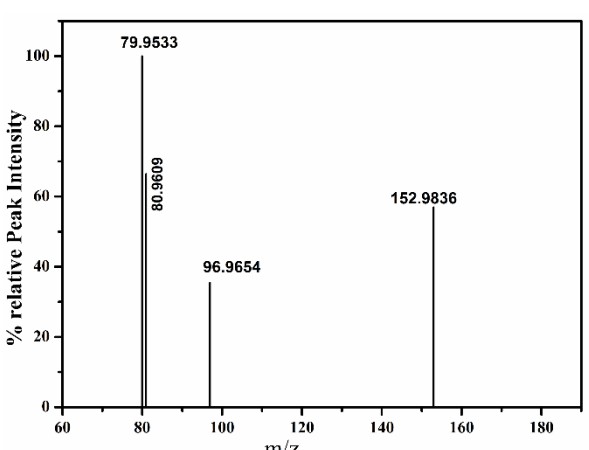


(d)

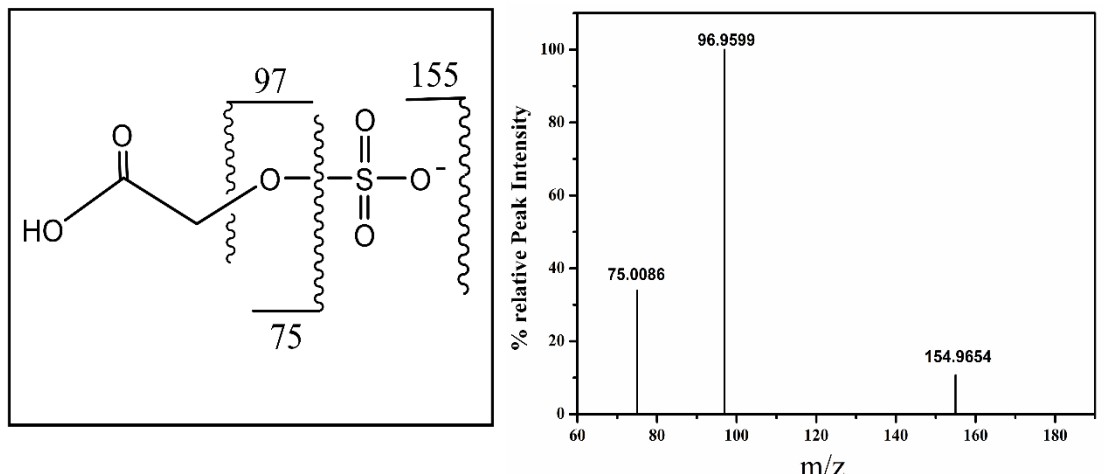


Figure 2. The fragmentation (left) and mass spectra (right) of benzyl sulfate (a), phenyl sulfate
(b), hydroxyacetone sulfate (c), and glycolic acid sulfate (d).

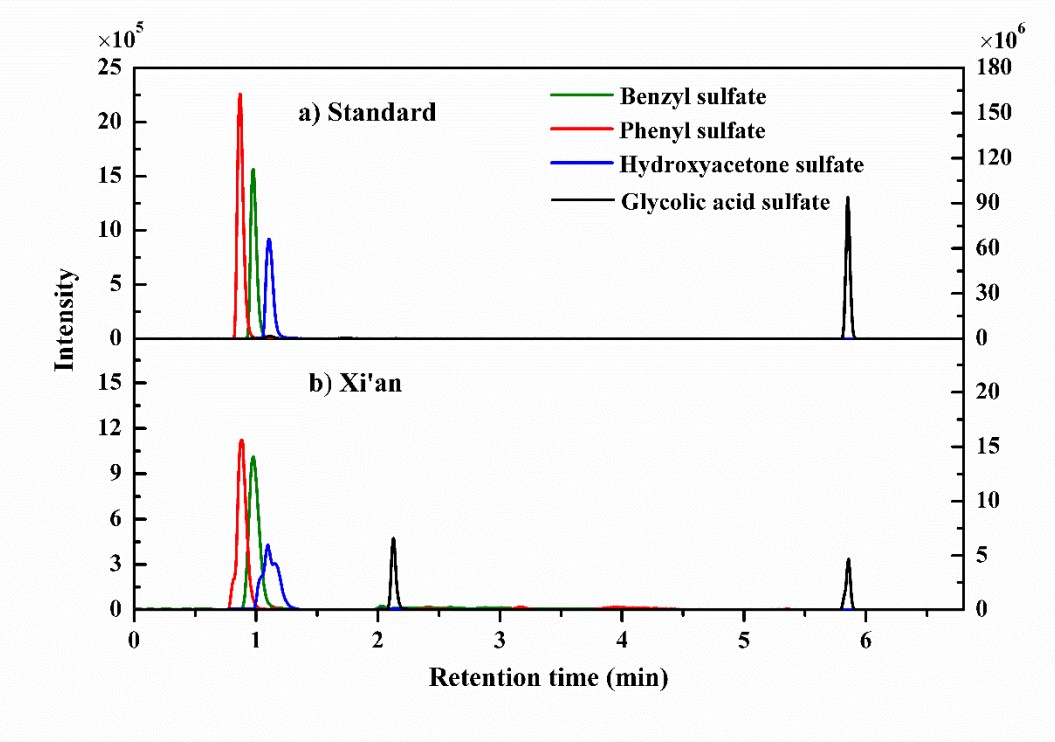


Figure 3. Typical chromatograms of organosulfates from the mixture of authentic standard
solution and ambient PM$_{2.5}$ samples, measured with the UPLC–ESI–MS/MS method. Note: the
intensity of benzyl sulfate and phenyl sulfate refers to the left Y axis and the intensity of
hydroxyacetone sulfate and glycolic acid sulfate refers to right Y axis.

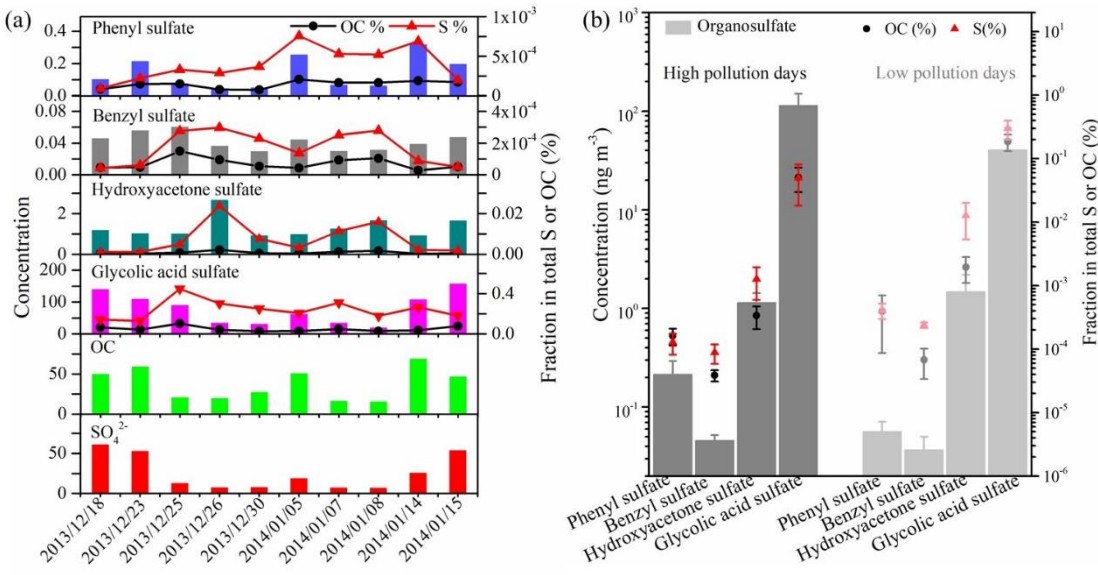

Figure 4. Time series of organosulfates (ng m$^{-3}$), OC (μg m$^{-3}$), SO$_4^{2-}$ (μg m$^{-3}$), and the fraction
of individual organosulfates in total sulfur and OC (a). The average concentrations of individual
organosulfates and the fractional contribution in total sulfur and OC during high and low
pollution days are also shown (b).