# Peer review of "Organosulfates in atmospheric aerosol: synthesis and quantitative analysis of PM$_{2.5}$ from Xi'an, Northwest China"

_Atmospheric Measurement Techniques, 2018_

## Referee Comment (RC1) · Anonymous Referee #1 · 3 May 2018

Identification and quantification of organosulfates are essential in assessing their formation pathway and contribution to particulate matter. A significant obstacle in the analysis of organosulfates is a lack of commercially available organosulfate standards. Nine organosulfates with different functional groups were synthesized in this study; and analysis of organosulfates on real ambient filters is subsequently conducted. The results are potentially interesting; the paper is very well written and current contribution is a welcome addition to the field.

Major comments:

1. Line 156: Any purification is performed to obtain the organosulfate standards?

2. Line 160: Are the synthesized organosulfate standards stable? Was there any degradation/decomposition observed during storage?

3. Line 163: Readers might be interested in the NMR spectra of the synthesized standards. Please include the information in SI.

4. Line 163-174: What is the purity of the obtained standards?

5. Line 313: The authors suggest the anthropogenic origin of phenyl sulfate and glycolic acid sulfate. Trend of phenyl sulfate is more similar to that of OC, while glycolic acid sulfate is more similar to inorganic sulfate. Any explanation?

6. Line 316-320: Do the authors suggest that hydroxyacetone sulfate is biogenic origin? If so, why the concentration of hydroxyacetone sulfate is much higher than that of phenyl and benzyl sulfate, which were suggested from anthropogenic origin? I doubt if there is a large biogenic source for hydroxyacetone sulfate in Winter.

Minor comments:

7. Line 322: Please define "high polluted days" and "low polluted days".

8. Line 615: "chromatogram" should read "chromatograms"

9. Line 611: Legends (a), (b), (c), (d) are missing on Figure 2

10. Line 620: Figure 4 (a), unit of concentration should be added to the legend.

---

## Referee Comment (RC2) · Anonymous Referee #2 · 7 May 2018

This manuscript reports the synthesis for nine organosulfates, including phenyl sulfate, 3-methylphenyl sulfate, benzyl sulfate, 2-methyl benzyl sulfate, 3-methyl benzyl sulfate, 2,4-dimethyl benzyl sulfate, 3,5-dimethyl benzyl sulfate, hydroxyacetone sulfate, and glycolic acid sulfate. Four standards were then used to optimize a UPLC-ESI-MS/MS method for identification and quantification of organosulfates. The novelty of this study lies in the application of the method to PM2.5 samples collected in urban air in Xi'an which is heavily polluted during winter, demonstrating the usefulness of the method. The authors primarily examined the potential of organosulfate formation only under wintertime conditions, which likely limits the influence of biogenic VOCs on organosulfate formation. The striking result of this study is the highest concentration of

glycolic acid sulfate in Xi'an, even higher than those reported in previous studies. This study suggests that glycolic acid sulfate is likely formed from anthropogenic VOCs in urban air in the presence of acidic sulfate aerosols.

Overall, the study is well done with all necessary details and the manuscript is well written. The authors provide a state-of-the-art overview of the knowledge in this field. The synthesis procedure of organosulfate standards is robust and well described. The UPLC-ESI-MS/MS is also well optimized. The organosulfate synthesis and the UPLC-MS method, though they are not completely new, are important and certainly adds to the quality of the field measurements. I recommend publication after minor revision.

Specific comments: 1. Organosulfate formation is very important under summer conditions, particularly due to the role of aqueous chemistry in producing sulfate which is essential for organosulfate formation. Why did the authors focus on winter conditions only? 2. In Figure 3, there is additional peak at RT of about 2.1 min in the ambient sample for glycolic acid sulfate? Please explain.

---

## Author Comment (AC1) · 2 Jun 2018

The authors thank the referee to review our manuscript and particularly for the valuable comments and suggestions that have significantly improved the manuscript. We have made most of the changes suggested by the referees and have outlined these in detail below.

Anonymous Referee #1

Identification and quantification of organosulfates are essential in assessing their formation pathway and contribution to particulate matter. A significant obstacle in the

analysis of organosulfates is a lack of commercially available organosulfate standards. Nine organosulfates with different functional groups were synthesized in this study; and analysis of organosulfates on real ambient filters is subsequently conducted. The results are potentially interesting; the paper is very well written and current contribution is a welcome addition to the field.

Major comments:

1. Line 156: Any purification is performed to obtain the organosulfate standards?

Response: The synthesized organosulfate standards were recrystallized in ethanol for purification. We have included this in revised manuscript.

2. Line 160: Are the synthesized organosulfate standards stable? Was there any degradation/decomposition observed during storage?

Response: The synthesized organosulfate standards were stored in refrigerator ($\sim$4 oC). No degradation/decomposition was observed after 2 years, which was confirmed by the NMR analysis. We have included this in revised manuscript.

3. Line 163: Readers might be interested in the NMR spectra of the synthesized standards. Please include the information in SI.

Response: Thanks for the suggestion, we have included the NMR spectra in SI.

4. Line 163-174: What is the purity of the obtained standards?

Response: The organosulfate standards were recrystallized in ethanol for purification and purity of these synthesized standards is ËČ95%, confirmed by NMR analysis. We have included this in revised manuscript.

5. Line 313: The authors suggest the anthropogenic origin of phenyl sulfate and glycolic acid sulfate. Trend of phenyl sulfate is more similar to that of OC, while glycolic acid sulfate is more similar to inorganic sulfate. Any explanation?

[Figure]

Response: Phenyl sulfate and glycolic acid sulfate both show a trend similar to OC and sulfate. However, we agree with the referee, there is small difference, which is likely associated with the difference in reaction rates, the abundance of precursors and their emission sources. Glycolic acid sulfate is formed efficiently from glycolic acid relative to glyoxal in the presence of acidic sulfate particles, therefore, sulfate is likely the reaction rate limited factor.

6. Line 316-320: Do the authors suggest that hydroxyacetone sulfate is biogenic origin? If so, why the concentration of hydroxyacetone sulfate is much higher than that of phenyl and benzyl sulfate, which were suggested from anthropogenic origin? I doubt if there is a large biogenic source for hydroxyacetone sulfate in Winter.

Response: We agree with the referee that emissions of biogenic precursors, e.g., isoprene, are not likely abundant in Xi'an in winter. In the manuscript, we discussed that in general hydroxyacetone sulfate has both biogenic and anthropogenic origins. It would depend on the formation rates of biogenic hydroxyacetone sulfate and anthropogenic hydroxyacetone sulfate. We have added more discussion in revised manuscript.

Minor comments:

7. Line 322: Please define "high polluted days" and "low polluted days".

Response: High polluted days with PM2.5 of 293.7-314.5 $\mu$g m-3 (average 300.6 $\mu$g m-3) and low polluted days with PM2.5 of 94.7-121.2 $\mu$g m-3 (average 106.4 $\mu$g m-3). We have added these in revised manuscript. We have included this in revised manuscript.

8. Line 615: "chromatogram" should read "chromatograms"

Response: Change made.

9. Line 611: Legends (a), (b), (c), (d) are missing on Figure 2

Response: Change made.

10. Line 620: Figure 4 (a), unit of concentration should be added to the legend.

Response: Because the unit for organosulfates (ng m-3) is different from that for OC and SO42- ($\mu$g m-3), we specify these in the caption.

Please also note the supplement to this comment:
https://www.atmos-meas-tech-discuss.net/amt-2018-104/amt-2018-104-AC1-supplement.pdf

––––––––––––––––––––––––––––

[Figure]

**Supplement:**

The $^1$H NMR and $^{13}$C NMR spectra of synthesized organosulfate standards are shown in Figure S1-S8 below. These $^1$H NMR and $^{13}$C NMR spectra were recorded on a Bruker Advance-III 400 MHz spectrometer at 400 and 100 MHz, respectively using trimethylsilane (TMS) as an internal standard.

[Figure]

Figure S1. $^1$H NMR spectrum of potassium phenyl sulfate in $D_2O$.

[Figure]

Figure S2. $^{13}$C NMR spectrum of potassium phenyl sulfate in D$_2$O.

[Figure]

Figure S3. $^1$H NMR spectrum of potassium benzyl sulfate in DMSO-*d*6.

[Figure]

Figure S4. $^{13}$C NMR spectrum of potassium benzyl sulfate in DMSO-*d*6.

[Figure]

Figure S5. $^{1}$H NMR spectrum of potassium hydroxyacetone sulfate in DMSO-*d*6.

[Figure]

Figure S6. $^{13}$C NMR spectrum of potassium hydroxyacetone sulfate in DMSO-*d*6.

[Figure]

[Figure]

Figure S7. $^{1}$H NMR spectrum of potassium glycolic acid sulfate in DMSO-*d*6.

[Figure]

Figure S8. $^{13}C$ NMR spectrum of potassium glycolic acid sulfate in $D_2O$.

---

## Author Comment (AC2) · 2 Jun 2018

The authors thank the referee to review our manuscript and particularly for the valuable comments and suggestions that have significantly improved the manuscript. We have made most of the changes suggested by the referees and have outlined these in detail below.

Anonymous Referee #2

This manuscript reports the synthesis for nine organosulfates, including phenyl sulfate, 3-methylphenyl sulfate, benzyl sulfate, 2-methyl benzyl sulfate, 3-methyl benzyl

sulfate, 2,4-dimethyl benzyl sulfate, 3,5-dimethyl benzyl sulfate, hydroxyacetone sulfate, and glycolic acid sulfate. Four standards were then used to optimize a UPLC-ESIMS/MS method for identification and quantification of organosulfates. The novelty of this study lies in the application of the method to PM2.5 samples collected in urban air in Xi'an which is heavily polluted during winter, demonstrating the usefulness of the method. The authors primarily examined the potential of organosulfate formation only under wintertime conditions, which likely limits the influence of biogenic VOCs on organosulfate formation. The striking result of this study is the highest concentration of glycolic acid sulfate in Xi'an, even higher than those reported in previous studies. This study suggests that glycolic acid sulfate is likely formed from anthropogenic VOCs in urban air in the presence of acidic sulfate aerosols. Overall, the study is well done with all necessary details and the manuscript is well written. The authors provide a state-of-the-art overview of the knowledge in this field. The synthesis procedure of organosulfate standards is robust and well described. The UPLC-ESI-MS/MS is also well optimized. The organosulfate synthesis and the UPLCMS method, though they are not completely new, are important and certainly adds to the quality of the field measurements. I recommend publication after minor revision.

Specific comments:

1. Organosulfate formation is very important under summer conditions, particularly due to the role of aqueous chemistry in producing sulfate which is essential for organosulfate formation. Why did the authors focus on winter conditions only?

Response: To the best of our knowledge, organosulfate related studies have been mainly associated with biogenic volatile organic compounds (VOCs), because of the large emissions of biogenic VOCs and the efficient aqueous chemistry for sulfate production in summer. Our understanding on the concentrations and formation mechanisms of organosulfates from anthropogenic VOCs is still very limited. During winter anthropogenic VOCs are very abundant in North China and aqueous-phase sulfate formation is also very efficient, particularly during haze period due to high relative hu-
midity. That is the motivation for us to focus on winter conditions.

2.In Figure 3, there is additional peak at RT of about 2.1 min in the ambient sample for glycolic acid sulfate? Please explain.

Response: In this study, we quantify organosulfates by monitoring a transition pair of precursor and product ions in the MRM mode of ESI-MS/MS analysis and by matching the retention time in UPLC analysis. When these two criteria are followed for ambient samples, the peaks can be assigned to a specific compound (i.e., organosulfate). The additional peak at RT of about 2.1 min in the ambient samples is not glycolic acid sulfate, because of different RT compared to the RT of glycolic acid standard. This peak is likely from a compound with same transition pair of ions as glycolic acid.